# OpenReview forum: "XOXO: Stealthy Cross-Origin Context Poisoning Attacks against AI Coding Assistants"
_ICLR.cc/2026/Conference — ICLR 2026 Conference Withdrawn Submission_

### Official Review · Reviewer_UCeW · 2025-10-27

**Soundness:** 2
**Presentation:** 3
**Contribution:** 3
**Rating:** 4
**Confidence:** 3

**Summary:**

This paper introduces "Cross-Origin Context Poisoning" (XOXO) , a novel attack exploiting how AI coding assistants automatically gather context from untrusted sources. Attackers poison shared code with semantics-preserving transformations (e.g., variable renaming) , which mislead the assistant's LLM into generating vulnerable code for a victim developer. The paper proposes a black-box algorithm, Greedy Cayley Graph Search (GCGS) , to help find these transformations more quickly by leveraging a "confidence monotonicity" property. The attack's effectiveness is demonstrated on code generation/vulnerability tasks and in a real-world demo against GitHub Copilot.

**Strengths:**

- **Novel and Practical Attack Vector**: The paper identifies an important and highly practical attack surface. The behavior of AI assistants automatically aggregating multi-origin context is indeed a potential security blind spot. The threat model presented (a malicious or compromised developer with commit privileges) is realistic, especially in the context of modern supply-chain attacks.
- **Concrete and Impactful Demonstration**: The end-to-end attack demonstration against GitHub Copilot is highly effective . Successfully inducing an SQL injection vulnerability through a subtle operation like changing a variable name from USE_RAW_QUERIES to RAW_QUERIES clearly demonstrates the real-world danger of this vulnerability.
- **Effectiveness in Vulnerability Injection**: The evaluation on CWEval is alarming. The research shows that this attack can not only generate functionally correct code but also inject security vulnerabilities simultaneously , successfully triggering 17 different CWE types. This indicates it can bypass the safety alignment mechanisms in modern LLMs.

**Weaknesses:**

- **Lack of Practicality for GCGS on Closed-Source Models**: The paper's core methodological contribution, the GCGS algorithm, has a major flaw regarding its applicability in the primary use case for AI coding assistants: closed-source models. Typically: a) The core mechanism of GCGS relies on model confidence scores (like perplexity, need to be calculated from extra information like logprobs). However, many mainstream closed-source models (like the tested Claude 3.5 Sonnet v2) do not provide this information via their API. b) The paper acknowledges this and suggests "prediction stability" as an alternative. However, this clearly requires "multiple runs" to estimate the confidence for a single input, which would multiply the query cost, making the attack prohibitively expensive and impractical against real-world closed-source models. This significantly limits the applicability of the paper's main contribution as it limits the attack scenarios significantly.
- **Insufficient and Contradictory Evaluation of Closed-Source Models**: As a direct consequence of the methodological flaw above, the paper's evaluation of closed-source models is weak and unconvincing. Typically: a) On Claude 3.5, the authors could not use GCGS and had to fall back to an "unguided" search, b) On GPT 4.1 (which does provide log probs), the data in Table 1 fails to demonstrate GCGS's efficiency advantage. On the MBPP+ task, GCGS (233 queries) was far less efficient than the "unguided" baseline on Claude 3.5 (75 queries). c) The paper lacks the most critical baseline: a direct comparison of GCGS vs. "unguided" search on the same closed-source model (e.g., GPT 4.1). d) The authors admit to only "one full run" for the main experiment due to cost, which makes the results lack statistical significance and further weakens the conclusions.
- **Insufficient Analysis of Defenses (Guardrails)**: The paper's discussion of defenses in Appendix E is qualitative and fails to provide sufficient insight. While the paper correctly points out that some defenses (like "human-in-the-loop") are impractical , it overlooks simple defenses tailored to this specific attack. The paper's implementation relies heavily on "identifier replacement" , and its most successful case (SQL injection) is based on a simple variable rename. Given this, an obvious defense would be a simple "keyword auditing" guardrail that detects or warns about suspicious, security-relevant identifier changes in the context. The paper fails to evaluate the effectiveness of such a simple, low-cost defense, making its discussion of countermeasures incomplete.

**Questions:**

- **GCGS Practicality and Efficiency**: How do you address the fact that GCGS's reliance on model confidence makes it inapplicable to many mainstream closed-source models like Claude 3.5? For models requiring "multiple runs" to estimate confidence, isn't the cost of GCGS prohibitively impractical? Furthermore, on GPT 4.1 where you could run GCGS, why was its query efficiency (e.g., 233 queries on MBPP+) so much worse than the "unguided" baseline on another model (75 queries)?
- **Defense Guardrails**: Your defense analysis in Appendix E focuses on complex or impractical solutions . However, given that your attack implementation (especially the GitHub Copilot demo) relies on simple identifier replacement , did you evaluate whether a simple "keyword auditing" guardrail (e.g., auditing security-relevant variable name changes in the context) could effectively mitigate or detect the XOXO attack?
- **Generalizability of Transformations**: Your current implementation appears to exclusively use identifier replacement . To what extent do you believe the attack's success is due to the GCGS algorithm itself, versus the specific effectiveness of "identifier replacement"? Have you attempted to apply GCGS to other semantics-preserving transformations (such as independent statement reordering), and how did it perform? Besides, I also wonder whether XOXO method can be extended to languages other than python and use transformations other than identifier replacement?

---

### Official Review · Reviewer_XE2p · 2025-10-29

**Soundness:** 3
**Presentation:** 3
**Contribution:** 3
**Rating:** 6
**Confidence:** 3

**Summary:**

This paper introduces a novel attack model named Cross-Origin Context Poisoning (XOXO), which targets AI coding assistants like GitHub Copilot. The attack manipulates code generation by subtly altering context, leading to vulnerabilities like SQL injection. The paper also proposes an algorithm, Greedy Cayley Graph Search (GCGS), to automate this attack by finding effective adversarial transformations.

**Strengths:**

- The XOXO attack is innovative and tackles a critical issue in AI-assisted software development, which is rarely addressed.
- The proposed GCGS algorithm is a valuable contribution, as it automates the identification of vulnerabilities without needing access to model gradients or prompt manipulation, making it applicable to proprietary systems.

**Weaknesses:**

- Given the popularity of GitHub Copilot, this paper uses it as an example to conduct a security analysis, which holds significant practical relevance. However, the subsequent experiments only employ LLMs to simulate code assistants, which seems to create a gap, as the simulation results may not adequately reflect the security issues present in real-world code assistants.
- The authors claim that the manipulations employed are semantics-preserving. However, based on the examples in the appendix, some function names have been replaced with entirely unrelated terms. Given the powerful comprehension capabilities of LLMs, it seems almost expected that such manipulations—which preserve functionality but alter semantics—could mislead the model.
- The threat model, which assumes a malicious developer with commit privileges, presents a somewhat narrow view of the security landscape. Given the prevalence of strict access controls in organizations, this internal threat scenario is less common in practice. Therefore, the real-world applicability of the attacks may be narrower.

**Questions:**

See Weaknesses.

---

### Official Review · Reviewer_NiVr · 2025-10-29

**Soundness:** 2
**Presentation:** 3
**Contribution:** 2
**Rating:** 2
**Confidence:** 4

**Summary:**

This paper proposes an inference-time attack method called XOXO, designed to perform context-level vulnerability injection against code generation models. In addition, it introduces a new black-box optimization algorithm, GCGS, which is based on a so-called confidence monotonicity property. GCGS operates by progressively reducing the model's confidence within a semantics-preserving transformation space, thereby inducing the model to produce incorrect or vulnerable code.
The method is evaluated on multiple open-source and closed-source models and reports high ASR on benchmarks such as HumanEval+, MBPP+, and CWEval.

**Strengths:**

1. The paper focuses on a practical and security-critical problem, namely context-based adversarial attacks on code generation models, which have significant implications for the safety of real-world developers.

2. The proposed GCGS algorithm is simple yet effective, allowing adversarial optimization under black-box conditions without requiring gradient information.

3. The experimental evaluation is comprehensive, covering both open-source and closed-source models, and reports metrics such as ASR, number of queries, and code similarity.

**Weaknesses:**

1. The definition of "Atomic Transformation" is unclear. The paper frequently uses this term but does not seem to explicitly specify which specific operations it includes.

2. CodeBLEU is not suitable for directly measuring naturalness. Although CodeBLEU effectively captures functional and structural similarity, it is insensitive to human-perceived readability. For instance, renaming a variable from `arr` to `VAR_1` or to an excessively long but valid name may keep syntax and dataflow unchanged, resulting in a high CodeBLEU score, while human readers would find the code unnatural [AR1].
[AR1] Yang, Chen, et al. "Dependency-aware code naturalness." Proceedings of the ACM on Programming Languages 8.OOPSLA2 (2024): 2355-2377.

3. CWEval results lack a clear unperturbed baseline. In the Section 5.2, the paper does not report the baseline ASR of CWEval without perturbations, making it unclear how much improvement XOXO actually provides. Additionally, the number of samples used for ASR computation and their coverage are not specified.

4. The trade-off between naturalness and attack success rate is not analyzed. The paper claims that GCGS balances naturalness and ASR but does not show their relationship. It would be interesting to know whether relaxing naturalness constraints (e.g., allowing larger-scale replacements) could significantly improve ASR, or whether strict naturalness requirements limit the attack's effectiveness. These questions deserve further exploration.

5. The difference between guided and unguided versions is small. The ASR gap between GCGS and the unguided baseline is relatively minor, while the number of queries required for CWEval is extremely high. This raises questions about the practical efficiency of the proposed guidance mechanism.

**Questions:**

1. Would it be better to use other approaches (such as human evaluation) as complementary metrics to CodeBLEU for measuring naturalness?

2. What is the actual improvement in XOXO's ASR, and how many samples were used in the evaluation?

3. If the naturalness constraint were relaxed, would XOXO's ASR increase significantly?

---

### Official Review · Reviewer_mAnM · 2025-11-01

**Soundness:** 2
**Presentation:** 3
**Contribution:** 3
**Rating:** 4
**Confidence:** 4

**Summary:**

The paper introduces Cross-Origin Context Poisoning (XOXO), a novel attack vector against AI coding assistants that automatically gather context from shared codebases.
The attack relies on semantics-preserving transformations (e.g., variable renaming) that alter context without changing program behavior.
The authors demonstrate that such benign transformations can mislead LLM-based coding assistants into generating insecure or incorrect code.
They further propose Greedy Cayley Graph Search (GCGS), a black-box algorithm that systematically searches for adversarial transformations by exploiting a newly identified monotonicity property in model confidence. The approach achieves high attack success rates (up to 83.67% on code generation and 66.67% on vulnerability injection tasks) across both open- and closed-source models.
The paper also demonstrates the attack on real-world coding assistant, tricking GitHub Copilot to generate insecure code.

**Strengths:**

- The introduction of GCGS is theoretically interesting. Modeling semantics-preserving transformations as a Cayley graph and leveraging monotonic confidence descent is an insightful conceptual contribution. The greedy search design fits the black-box attack setting well.
- The experiments span multiple families of code LLMs, both open and proprietary, and cover diverse tasks: code generation, secure coding, and reasoning.
- The end-to-end Copilot attack example grounds the work in a real-world context and effectively conveys the practical implications of the threat.

**Weaknesses:**

- The assumed threat model presumes an attacker with commit privileges and detailed knowledge of the victim’s assistant configuration and prompt templates. While realistic for insider or supply-chain threats, it represents a relatively strong assumption that may limit the general applicability of the attack scenario in less tightly coupled development environments.
- The number of model queries required for the vulnerability injection attack is significantly higher than for standard code generation tasks, which is often in the thousands. This raises concerns about the attack’s practicality.
- GCGS's empirical gains over the no guidence baseline in both success rate and query efficiency are modest and often within reported variance ranges.

**Questions:**

- Given that the attack requires a large number of model queries, how stable are the optimized adversarial contexts under realistic inference settings (e.g., non-zero and higher temperature)? To what extent could the observed misbehavior be attributed to sampling randomness rather than the crafted context itself?
- Are there any potential detection or defense mechanisms you foresee for mitigating the attacks? How did the vendor respond to the attack against Copilot?
- What other types of transformations beyond variable renaming are used? More detailed examples would help clarify the range and naturalness of the perturbations used in the attack.

---

### Note · Authors · 2025-11-20

I have read and agree with the venue's withdrawal policy on behalf of myself and my co-authors.